# Effects of Therapist Intervention during Upper-Extremity Robotic Rehabilitation in Patients with Stroke

**DOI:** 10.3390/healthcare11101369

**Published:** 2023-05-10

**Authors:** Si-Yun Kim, Yu-Mi Kim, See-Won Koo, Hyun-Bin Park, Yong-Soon Yoon

**Affiliations:** Department of Rehabilitation Medicine, Presbyterian Medical Center, Jeonju 54987, Republic of Korea; sazure@naver.com (S.-Y.K.);

**Keywords:** stroke, upper-extremity, robot rehabilitation, therapist intervention

## Abstract

This study aimed to determine whether the treatment effect differs for patients with stroke who perform robot-assisted upper-extremity rehabilitation by themselves compared to those whose rehabilitation is actively assisted by a therapist. Stroke patients with hemiplegia were randomly divided into two groups and received robot-assisted upper-limb rehabilitation for four weeks. In the experimental group, a therapist actively intervened in the treatment, while in the control group, the therapist only observed. After four weeks of rehabilitation, the manual muscle strength, Brunnstrom stage, Fugl-Meyer assessment of the upper-extremity (FMA-UE), box and block test, and functional independence measure (FIM) showed significant improvement in both groups compared to that before treatment; however, no interval change in spasticity was noted. The post-treatment values showed that the FMA-UE and box and block tests were significantly improved in the experimental group compared to those in the control group. Comparing the changes in the pre- and post-treatment values, the FMA-UE, box and block test, and FIM of the experimental group were significantly improved compared to those in the control group. Our results suggest that active intervention by therapists during robot-assisted upper-limb rehabilitation positively impacts upper-extremity function outcomes in patients with stroke.

## 1. Introduction

The decreased function of the hemiplegic upper-extremity after a stroke restricts activities of daily life and social participation in patients with stroke, thus reducing their quality of life [1,2,3,4,5]. Therefore, various treatments are being attempted to improve upper-limb function in patients with stroke. Based on learnings from neuroplasticity, task-oriented rehabilitation training is effective, and the higher the intensity, the better the activities of daily life [6,7]. For this reason, upper-limb robotic rehabilitation has attracted attention because it is a safe and efficient way to perform repetitive training while performing task-oriented rehabilitation after a stroke [8]. Upper-extremity rehabilitation treatment using a robot is more effective in improving motor function than traditional upper-extremity rehabilitation treatment in patients with chronic stroke [9]. In patients with subacute stroke, robotic rehabilitation therapy has similar effects as traditional therapy, but its superiority has not been demonstrated [10]. However, it has advantages such as low cost [11], convenience, and increased patient compliance while achieving similar effects. The use of therapy personnel has also been found to be efficient [12]. This is because if the therapist helps the patient set up the initial settings, patients can perform the task by themselves using the visual and auditory information provided by the screen and sensor on the control panel. Even though upper-extremity rehabilitation using robots has been introduced for some time, no clear protocol exists for it [12]. Although previous studies have compared the effects of traditional upper-limb rehabilitation and robot-assisted upper-limb rehabilitation, there is no study on how a therapist’s intervention affects robot-assisted upper-limb rehabilitation. Therefore, this study aimed to determine whether the treatment effect is different in patients with stroke who perform robot-assisted upper-limb rehabilitation by themselves compared to those whose rehabilitation is actively assisted by a therapist.

## 2. Materials and Methods

### 2.1. Design and Sample Size

This study is a prospective, randomized controlled trial. Patients were randomly divided into two groups and received robot-assisted upper-limb rehabilitation for four weeks. In the experimental group, a therapist actively intervened during the treatment, while in the control group, a therapist only observed. Both groups underwent pre- and post-treatment evaluations (Figure 1). Randomization was processed using a centrally-generated, variable-sized block design made using a protected and concealed statistic program.

The sample size was estimated using the formula of ‘normality test for two independent means in continuous results. Referring to the previous study investigating functional recovery after stroke [13], the standard deviation value of functional independence measure (FIM) at admission and the change in FIM mean value between admission and discharge were applied to the formula. When applying the data to the formula, a significance level of 0.05 and power of 0.8 required a minimum sample size of each group of *n* = 15.68. We enrolled a total of 38 patients, 19 in each group, exceeding the minimum recommended number.

### 2.2. Patients

Patients admitted to the Department of Rehabilitation Medicine at Jesus Hospital were enrolled between March 2020 and February 2021. Thirty-eight hemiplegic patients aged ≥ 18 years diagnosed with cerebral hemorrhage or cerebral infarction using brain magnetic resonance imaging or computerized tomography with more than 2 weeks of disease onset were selected. The patients’ manual muscle strength at the elbow joint was grade 2 or higher and the spasticity of the hemiplegic shoulder and elbow joint ranged between zero and three on the modified Ashworth scale (MAS). Patients with a history of damage to the paralyzed upper-limb, those with difficulty understanding simple instructions or action presentations due to severe aphasia, and those whose neurological and functional recovery was not inspected medically were excluded. Patients with peripheral nerve damage or peripheral neuropathy, serious mental illness, pregnancy or breastfeeding, a genetic history of cerebral infarction induction, or a history of hyper thrombosis were also excluded. This study was approved by the Jesus Hospital Clinical Research Ethics Review Board (approval number 2012-11-52). 

### 2.3. Robot Device

Upper-extremity rehabilitation robots can be largely divided into end-effector-based and exoskeleton-type robots. When using end-effector-based robots, the patient’s arm and a part of the machine are connected and can move the upper-limb within a set ROM [14]. Exoskeletal is a form of direct control of each joint by forming an axis according to the anatomical alignment of the upper-limb [11]. In a study by Chang et al. in 2013, the effect of an exoskeletal robot on improving upper-extremity motor skills in patients with stroke was found to be insufficient [15]. In a 2020 meta-analysis by Mehrholz et al., the type of robot did not significantly affect the treatment effect [16]. Neuro-X is an easy-to-use, end-effector-based robot that is small, economical, and commercially available, and is effective in improving upper-limb function and cognitive function in patients with chronic stroke [17]. Therefore, we used Neuro-X for the robot-assisted rehabilitation of upper-limb in this study (Figure 2).

### 2.4. Interventions

Treatment was performed for 4 weeks, 35 min per day, 5 days per week using an upper-extremity rehabilitation robot (Figure 2). Continuous passive motion (CPM) exercises (Figure 3A), isometric task-specific exercises (Figure 3B), range of motion continuous active motion (ROM CAM) exercises (Figure 3C), and 360° continuous active motion (360° CAM) exercises (Figure 3D) were performed. In both the experimental and control groups, the therapist helped the patient to be seated correctly before treatment, raised and fixed the hemiplegic upper-limb on the robot control panel, and taught the patient how to perform the task. When the robot training started in the experimental group, verbal and visual instructions using a stick were given if the therapist thought there was a problem with the patient’s treatment performance. We also allowed encouragement other than objective instructions. For example, when the patient loses concentration during training, the therapist points the stick at the screen or gives verbal instructions to induce concentration. In addition, when the patient was slow to perform the program, the therapist helped by using a stick to remind them of the method. In contrast, in the control group, the therapist observed the patients without any intervention.

### 2.5. Clinical Assessments

Pre- and post-treatment evaluations were performed to measure the neurological recovery stage, degree of muscle tone and strength, and upper-extremity function. The evaluations included the Brunnstrom stage, MAS, manual muscle test (MMT), Fugl-Meyer assessment of the upper-extremity (FMA-UE), box and block test, and functional independence measure (FIM). The neurological recovery stage was evaluated through the Brunnstrom recovery stage. Shoulder flexion, elbow flexion, and wrist extension on the hemiplegic side were evaluated using manual muscle strength evaluation. Grip power and pinch power were measured using a measuring system, and spasticity of the shoulder and elbow joints was measured using MAS. FMA-UE and box and block tests were performed for upper-extremity function evaluation. Moreover, the FIM was performed to evaluate the overall function in activities of daily life.

#### 2.5.1. Brunnstrom Stage

It is a method to evaluate the degree of recovery of motor function in six stages according to muscle tone, muscle strength, and synergy after a stroke. Stage 1 refers to a state with almost no muscle tone, and stage 6 refers to a state of almost normal movement [13].

#### 2.5.2. Manual Muscle Strength Evaluation

This test measures the ability of a single muscle or specific muscles acting on a specific joint to contract voluntarily. It is evaluated from level 0, with no muscle contraction, to level 5, where active normal joint movement is possible under gravity and sufficient resistance. In stage 3, active joint movement is possible by overcoming gravity [19].

#### 2.5.3. MAS

In this method, the joint is passively flexed or extended over the full ROM when resting, and the degree of the resistance felt by the examiner is expressed on a five-level ranking scale (MAS 0–5). MAS 0 indicates no hypertonia, and MAS 4 indicates that exercise is impossible in the flexion or extension position owing to high muscle tone [20].

#### 2.5.4. FMA-UE

It is used to evaluate the upper-limb motor function and nerve recovery degree of patients with stroke. It is divided into four categories, and a total of 33 items are evaluated on a three-point scale, with a perfect score being 66 points; complete assistance or no muscle reflex response is scored as 0 points, partial assistance required to complete the movement is scored as 1 point, and appropriate completion of the movement is scored as 2 points [21].

#### 2.5.5. Box and Block Test

To evaluate the upper-extremity gross motor control ability, 150 cube blocks, each 2.5 cm in size, and a rectangular wooden box with a partition in the center were used. The goal was to move the maximum number of cubes, one by one, from one box compartment to the other within 60 s [22].

#### 2.5.6. FIM

It consists of six items, four for motor ability and two for cognitive ability, and is divided into 18 detailed categories of motion. In addition, a seven-level score is given according to the level of performance. A score of 18 is given if the patient needs complete assistance in performing all items, and a score of 126 is obtained if the patient can perform all items independently [23].

### 2.6. Statistical Analysis

The homogeneity test for demographic and general characteristics between the experimental and control groups was compared using the χ2 (chi-square) or the Mann–Whitney tests. The significance level was set at *p* < 0.05. Functional parameters before and after treatment in the same group were evaluated using the Wilcoxon signed-rank test, and the significance level was set at *p* < 0.05. Changes before and after the start of treatment in the experimental and control groups were evaluated using the Mann–Whitney test, and the significance level was set at *p* < 0.05.

## 3. Results

### 3.1. General Characteristics

Thirty-eight stroke patients who participated after agreeing to the purpose of the study were randomly assigned and divided into two groups: experimental (19 participants) and control (19 participants). The experimental group had an average age of 60.74 (11.84) years, with thirteen male and six female patients, and the control group had an average age of 63.42 (6.07) years, with twelve male and seven female patients. When the diagnosis, lesion type, location, and disease duration were confirmed, in the experimental group, there were eleven patients with right hemiplegia, eight with left hemiplegia, seven with cerebral infarction, twelve with cerebral hemorrhage, six with cortical lesions, and nine with subcortical lesions. Four patients had cortical and subcortical lesions, and the mean duration of disease was 13.7 9(13.48) months. In the control group, there were ten patients with right hemiplegia, nine with left hemiplegia, nine with cerebral infarction, ten with cerebral hemorrhage, five with cortical lesions, nine with subcortical lesions, and five with both cortical and subcortical lesions. The mean disease duration was 12.21 (10.40) months. The average Mini-Mental State Examination score was 22.13 (7.28) points in the experimental group and 21.70 (6.73) points in the control group. 

When the general characteristics of the two groups were compared, no significant differences were observed between the two groups regarding age, sex, the direction of hemiplegia, type and location of the lesion, duration of disease, and cognitive function (*p* > 0.05) (Table 1). Thus, it was possible to exclude bias in determining the effect of the treatment.

### 3.2. Function of Hemiplegic Upper-Extremity

#### 3.2.1. Upper-Extremity Function before Treatment

Before treatment, no statistically significant differences were observed between the two groups in the MMT, MAS, Brunnstrom stage, FMA-UE, the box and block test, 10-s test, and FIM. It was possible to exclude bias in determining the effect of the treatment (*p* > 0.05) (Table 2).

#### 3.2.2. Upper-Extremity Function after Treatment

When the upper-extremity function was compared and evaluated between the experimental and control groups after treatment, the experimental group showed more improvement than the control group in the FMA-UE and box and block tests. The FMA-UE score was 44.37 points in the experimental group and 39.47 points in the control group (*p* = 0.048), and the box and block test score was 23.74 points in the experimental group and 19.95 points in the control group, showing statistically significant differences (*p* = 0.042) (Table 2). However, there was no difference between the two groups in the MMT, spasticity, Brunnstrom stage, or FIM (*p* > 0.05).

#### 3.2.3. Changes in Upper-Extremity Function before and after Treatment

After four weeks of treatment, the experimental and control groups showed significant improvements in the MMT, Brunnstrom stage, FMA-UE, box and block test, and FIM after treatment. However, spasticity was not improved in either group (*p* > 0.05).

When the upper-limb manual muscle strength was evaluated after four weeks of rehabilitation, the experimental and control groups showed statistically significant improvements in shoulder flexion, elbow flexion, wrist extension, gripping, and pinching strength. In the experimental group, the shoulder flexor strength significantly improved from 2.21 before treatment to 2.68 after treatment (*p* = 0.011), and the elbow flexor strength significantly improved from 2.35 before treatment to 2.68 after treatment (*p* = 0.020). The wrist extensor strength significantly improved from 2.26 before treatment to 2.63 after treatment (*p* = 0.011). The grip power improved from 15.00 before treatment to 24.89 after treatment (*p* = 0.001), and the pinch power improved from 7.05 before treatment to 9.66 after treatment (*p* = 0.006). In the control group, the shoulder flexor strength significantly improved from 2.32 before treatment to 2.74 after treatment (*p* = 0.005), and the elbow flexor strength significantly improved from 2.34 before treatment to 2.84 after treatment (*p* = 0.020). The wrist extensor strength significantly improved from 2.02 before treatment to 2.47 after treatment (*p* = 0.011). The grip power improved from 16.16 before treatment to 24.79 after treatment (*p* = 0.000), and the pinch power improved from 7.10 before treatment to 9.34 after treatment (*p* = 0.015).

The MAS score showed no significant changes before and after treatment in both the experimental and control groups (*p* > 0.05). 

The Brunnstrom stage significantly improved from 2.84 before treatment to 3.95 after treatment in the experimental group (*p* = 0.005). In the control group, the Brunnstrom stage also significantly improved from 2.94 before treatment to 3.63 after treatment (*p* = 0.010). 

The FMA-UE showed a statistically significant improvement, from 35.05 points before treatment to 44.37 points after treatment in the experimental group (*p* = 0.001). In the control group, the FMA-UE also significantly improved from 34.95 points before treatment to 39.47 points after treatment (*p* = 0.001).

The box and block test showed a significant improvement in the experimental group, from 15.68 points before treatment to 23.74 points after treatment (*p* = 0.001). In the control group, there was a significant improvement from 15.95 points before treatment to 19.95 points after treatment (*p* = 0.003). 

The FIM showed a significant improvement from 69.26 points before treatment to 87.84 points after treatment in the experimental group (*p* = 0.001) and a significant improvement from 69.05 points before treatment to 78.05 points after treatment in the control group (*p* = 0.001) (Table 2).

#### 3.2.4. Comparison of Changes before and after Treatment between Each Group

Compared to the control group, the experimental group showed more improvement in the FMA-UE, box and block test, and FIM (Table 3, Figure 4). However, there was no difference between the two groups in the MMT, spasticity, or Brunnstrom stage (*p* < 0.05) (Table 3). 

The change in the FMA-UE before and after treatment was 9.32 (5.26) points in the experimental group and 4.53 (4.90) points in the control group (*p* = 0.045), and the change in the box and block test scores before and after treatment was 8.05 (5.68) in the experimental group and 4.00 (4.99) in the control group (*p* = 0.015). The change in the FIM was 18.58 (9.83) points in the experimental group and 9.00 (6.00) points in the control group (*p* = 0.003). The differences in all three items (FMA-UE, box and block test, and FIM) were statistically significant.

The change in the MMT grade of the shoulder flexor strength was 0.47 (0.77) in the experimental group and 0.42 (0.51) in the control group (*p* = 0.851). The change in the MMT grade of the elbow flexor strength was 0.37 (0.60) in the experimental group and 0.47 (0.51) in the control group (*p* = 0.421). The change in the MMT grade of the wrist extensor strength was measured at 0.37 (0.60) in the experimental group and 0.42 (0.61) in the control group (*p* = 0.752). The change in grip power was 9.89 (11.57) in the experimental group and 8.63 (8.86) in the control group (*p* = 0.895), and the change in pinch power was measured at 2.61 (3.61) in the experimental group and 2.24 (3.65) in the control group (*p* = 0.656).

The change in MAS at the elbow was −0.05 (0.85) in the experimental group and −0.11 (1.05) in the control group (*p* = 0.924); at the wrist, −0.16 (0.90) in the experimental group and −0.05 (0.71) in the control group (*p* = 0.631); at the finger, 0.11 (0.94) in the experimental group and −0.16 (0.90) in the control group (*p* = 0.458). There was no statistically significant difference.

The change in the Brunnstrom stage was 1.11 (1.29) in the experimental group and 0.68 (0.95) in the control group, with no significant difference (*p* = 0.340).

#### 3.2.5. Detailed Classification of FMA-UE

The detailed classification of the FMA-UE in movement includes the upper-extremity reflex and ROM of the shoulder: category A, which evaluates the nerve recovery stage; and category B, which evaluates a joint ROM and movement of the wrist, flexion and extension of the hand and fingers, and range of joint motion. It is divided into two parts: category C, which evaluates movement and coordination; and category D, which evaluates coordination and speed. When comparing the amount of change before and after treatment for each subcategory in the experimental and control groups, there was no significant difference between the two groups (Table 4).

## 4. Discussion

It is known that robot-assisted rehabilitation treatment to restore upper-limb function in patients with stroke can safely and efficiently increase the intensity and number of repetitions of rehabilitation treatment. A 2020 study by Ranzani et al. reported that the effects of robot-assisted upper-extremity rehabilitation treatment in patients with subacute stroke were not inferior to traditional occupational therapy and showed significant improvement in upper-extremity muscle strength and function [24]. This study showed no significant difference between the experimental and control groups during the pre-treatment evaluation. Nevertheless, the evaluation values after treatment increased effectively, indicating that treatment using the upper-extremity rehabilitation robot effectively improves upper-extremity muscle strength, nerve recovery stage, and upper-extremity function in patients with subacute stroke, regardless of intervention by the therapist. Although some studies have reported significant improvement in the spasticity of the hemiplegic side through upper-extremity rehabilitation robots [25,26], reports of no change or even worse results have also been published [6,7].

In this study, there was no significant change in the spasticity of the shoulders, arms, wrists, and fingers in both groups before and after treatment; therefore, in this study, the effect of the therapist’s intervention and upper-limb-assisted robot rehabilitation treatment on the improvement of spasticity is not clear. Spasticity is mainly managed with oral medications and interventional procedures, including botulinum toxin injections [27], and stretching is widely used to reduce post-stroke spasticity. Stretching may help improve post-stroke spasticity, but there is no conclusive evidence on the effectiveness of stretching interventions for improving spasticity [28]. Furthermore, it is thought that the upper-limb-assisted robot used in this study has little stretching effect, so there was no improvement in spasticity.

When comparing the evaluation items of the two groups before and after treatment, the FMA-UE and box and block evaluations showed significant improvement in the experimental group, which can be interpreted as the therapist’s intervention having enhanced the improvement of the upper-extremity function in the patients undergoing robotic rehabilitation. The improvements in the box and block evaluation, a single-item assessment, can be interpreted as the therapist’s intervention during robot-assisted upper-extremity rehabilitation treatment contributing to improving the hemiplegic side’s gross motor function after stroke.

The FMA-UE was a multi-item evaluation, and the total showed a significant improvement in the change before and after treatment. However, there was no significant difference when dividing it into four subcategories. It was difficult to reach statistical significance because the number of people evaluated was small when divided into subcategories. The detailed evaluation items indicate that the improvements in the joint ROM, movement control, and speed may be related to the therapist’s intervention. However, a follow-up study with a sufficient number of patients is required.

A stroke causes damage to the brain’s neurotransmitter system, resulting in reduced arousal and attention [29]. The upper-extremity rehabilitation robot provides various programs, visual effects, vibrations, and sounds to improve the patient’s arousal and performance; however, over time, the patient learns the repeated stimulation within the set range, and the treatment effect may be decreased.

At this time, the therapist’s active intervention and encouragement can compensate for this. Therefore, it yields better treatment effects than the group without the therapist’s intervention. However, further studies are required as this study did not evaluate the patient’s arousal and concentration during treatment. The limitation of this study is that the intervention was performed under the therapist’s judgment, and a specific manual for implementing the intervention was not provided. In the future, obtaining quantitative data by designing a study including a manual will be possible.

The study was conducted on patients who had been diagnosed with stroke for more than two weeks. Sixteen patients had less than six months of onset and 22 patients had more than six months of onset. The mean duration of onset was 12 to 13 months (Table 1). A follow-up study is needed to determine whether the effect of robot-assisted upper-extremity rehabilitation varies depending on the onset period and whether the effect of the therapist’s intervention changes.

## 5. Conclusions

In this study, during rehabilitation treatment using an upper-limb robot for patients with stroke, it was found that the upper-limb function improved more significantly when there was active intervention by the therapist than when there was no intervention. These results will be helpful when developing guidelines for using upper-limb-assisted rehabilitation robots or when developing robot rehabilitation feedback programs in the future.

## Figures and Tables

**Figure 1 healthcare-11-01369-f001:**
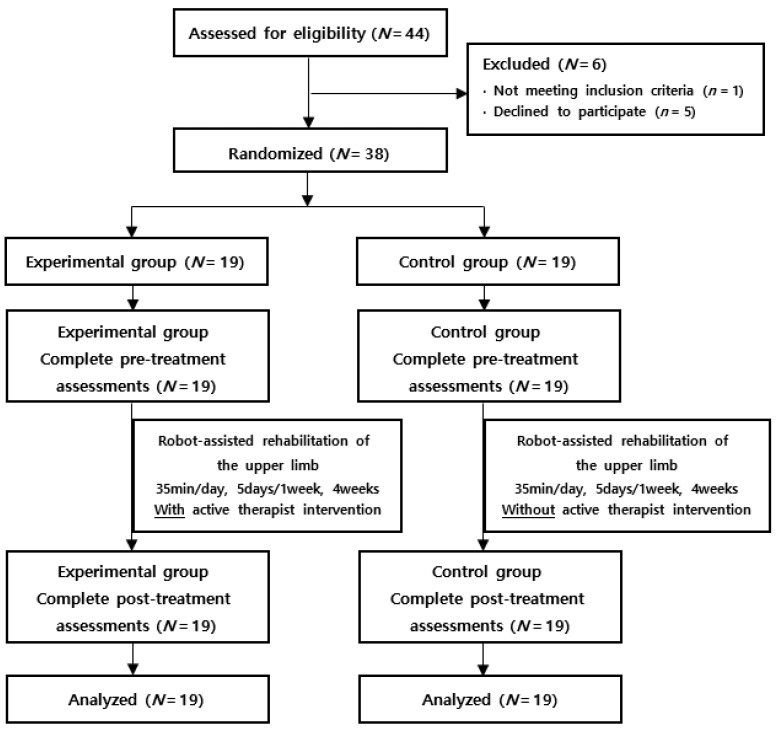
Flow diagram of the study.

**Figure 2 healthcare-11-01369-f002:**
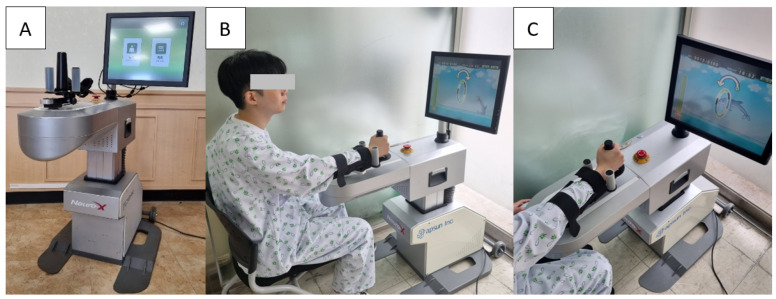
Robot-assisted upper-extremity training system (Neuro-X; Apsun, South Korea). (**A**). A 17-inch color touch screen monitor, base for upper-extremity exercise, and main body. (**B**,**C**). Patient sitting on a chair and training the paralyzed upper-extremity.

**Figure 3 healthcare-11-01369-f003:**
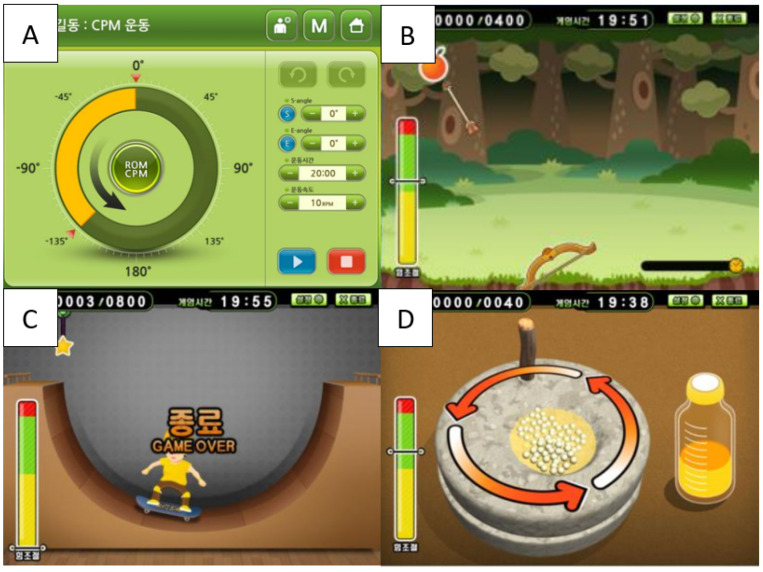
Programs of robot-assisted upper-extremity training system [18]. (**A**). Continuous passive motion (CPM): program to passively rotate the handle 360° or move it within a set angle range. (**B**). Isometric task specific exercise: program to perform upper-extremity isometric exercise with the armrests locked. (**C**). Range of motion continuous active motion (ROM CAM) exercise: program to actively move the upper-limb within a set angle range. (**D**). 360° continuous active motion (CAM) exercise: program to actively move the handle while rotating 360°.

**Figure 4 healthcare-11-01369-f004:**
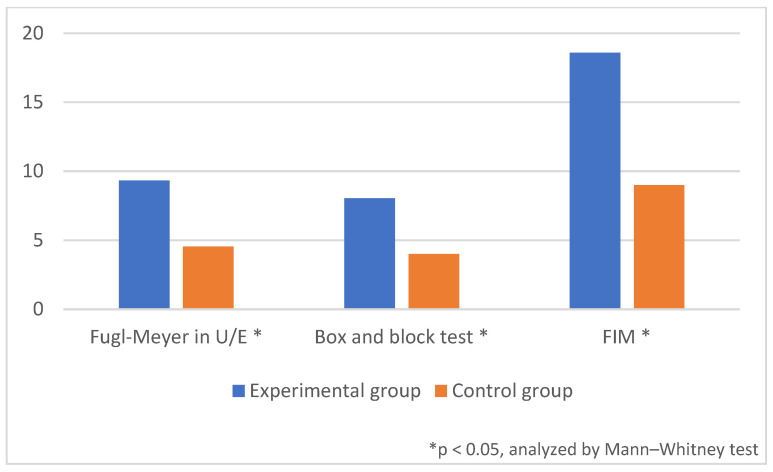
Comparison of the difference in measures before and after treatment: Fugl-Meyer in upper-extremities, box and block test, and functional independence measure (FIM). * *p* < 0.05, analyzed by Mann–Whitney test.

**Table 1 healthcare-11-01369-t001:** General characteristics and baseline functional variables in patients.

Variables	Experimental Group	Control Group	Mean	*p*
Number of participants	19	19	19	NA
Age (years)	60.74 (11.84)	63.42 (06.70)	62.08 (10.63)	0.801
Sex				0.652
Male	13	12	12.5	
Female	6	7	6.5	
Diagnosis (Number)				0.803
Right hemiplegia	11	10	10.5	
Left hemiplegia	8	9	8.5	
Type of lesion (Number)				0.591
Infarction	7	9	8	
Hemorrhage	12	10	11	
Site of lesion (Number)				0.612
Cortex	6	5	5.5	
Subcortex	9	9	9.0	
Both (cortex and subcortex)	4	5	4.5	
Duration (month)	13.79 (13.48)	12.21 (10.40)	13.00 (11.90)	0.827
BMI	20.63 (3.75)	21.26 (3.60)	20.94 (3.64)	0.600
MMSE	22.13 (7.28)	21.70 (6.16)	21.92 (6.73)	0.764

Mean (S.D), The data were analyzed by Mann–Whitney test and chi-square test. NA: Not Applicable. MMSE: Mini-Mental State Examination.

**Table 2 healthcare-11-01369-t002:** Motor functions and activity of daily living assessments.

Variables	Experimental Group		Control Group	
	Pre-Treatment	Post-Treatment	Pre-Treatment	Post-Treatment
MMT				
Shoulder flexion	2.21 (0.98)	2.68 (0.82) *	2.32 (0.67)	2.74 (0.56) *
Elbow flexion	2.35 (0.94)	2.68 (0.82) *	2.34 (0.68)	2.84 (0.60) *
Wrist extension	2.26 (1.19)	2.63 (1.12) *	2.02 (0.91)	2.47 (0.84) *
Grip power	15.00 (18.00)	24.89 (23.43) *	16.16 (17.21)	24.79 (22.71) *
Pinch power	7.05 (6.64)	9.66 (6.65) *	7.10 (5.67)	9.34 (6.19) *
MAS				
Shoulder	1.10 (1.12)	1.00 (1.00)	1.10 (0.87)	0.95 (0.91)
Elbow	1.16 (1.12)	1.10 (1.15)	1.05 (0.97)	0.95 (0.91)
Wrist	1.12 (1.08)	1.05 (1.08)	1.00 (1.20)	0.95 (0.91)
Finger	1.00 (1.20)	1.11 (1.15)	1.11 (0.88)	0.95 (0.91)
Brunnstrom stage	2.84 (1.86)	3.95 (1.27) *	2.94 (1.47)	3.63 (1.12) *
Fugl-Meyer in U/E	35.05 (25.82)	44.37 (22.10) *†	34.95 (22.72)	39.47 (24.16) *
Box and block test	15.68 (18.69)	23.74 (16.46) *†	15.95 (17.67)	19.95 (19.09) *
FIM	69.26 (27.17)	87.84 (24.64) *	69.05 (27.99)	78.05 (28.05) *

* *p* < 0.05, between pre- and post-treatment in the same group analyzed by Wilcoxon signed-rank test. † *p* < 0.05, between post-treatment in the experimental group and post-treatment in the control. group analyzed by Mann–Whitney test. MMT: Manual Muscle Test. MAS: Modified Ashworth Scale. FIM: Functional Independence Measure. U/E: Upper-extremity.

**Table 3 healthcare-11-01369-t003:** Comparison of changes of clinical assessments before and after treatment.

Variables	Experimental Group	Control Group	Mean	*p*
MMT				
Shoulder flexion	0.47 (0.77)	0.42 (0.51)	0.45 (0.64)	0.851
Elbow flexion	0.37 (0.60)	0.47 (0.51)	0.42 (0.55)	0.421
Wrist extension	0.37 (0.60)	0.42 (0.61)	0.39 (0.59)	0.752
Grip power	9.89 (11.57)	8.63 (8.86)	9.26 (10.18)	0.895
Pinch power	2.61 (3.61)	2.24 (3.65)	2.42 (3.58)	0.656
MAS				
Elbow	−0.05 (0.85)	−0.11 (1.05)	−0.08 (0.94)	0.924
Wrist	−0.16 (0.90)	−0.05 (0.71)	−0.10 (0.80)	0.631
Finger	0.11 (0.94)	−0.16 (0.90)	−0.03 (0.91)	0.458
Brunnstrom stage	1.11 (1.29)	0.68 (0.95)	0.89 (1.13)	0.340
Fugl-Meyer in U/E	9.32 (5.26)	4.53 (4.90)	6.92 (11.44)	0.045 *
Box and block test	8.05 (5.68)	4.00 (4.99)	6.03 (5.66)	0.015 *
FIM	18.58 (9.83)	9.00 (6.00)	13.79 (9.39)	0.003 *

Mean (S.D). * *p* < 0.05, analyzed by Mann–Whitney test. MMT: Manual Muscle Test. MAS: Modified. Ashworth Scale. FIM: Functional Independence Measure. U/E: Upper-extremity.

**Table 4 healthcare-11-01369-t004:** Comparison of changes before and after treatment for each subcategory of the FMA-UE.

Variables	Experimental Group	Control Group	Mean	*p*
Category A	4.50 (9.51)	2.33 (4.19)	3.42 (7.32)	0.767
Category B	1.72 (2.49)	1.11 (0.96)	1.42 (1.89)	0.696
Category C	2.28 (3.37)	1.61 (1.72)	1.94 (2.66)	0.938
Category D	0.61 (1.14)	0.33 (0.59)	0.47 (0.91)	0.913

Mean (S.D).

## Data Availability

Data will be available upon reasonable request to the authors.

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
