# Peer review of "Effects of Therapist Intervention during Upper-Extremity Robotic Rehabilitation in Patients with Stroke"

_healthcare, 2023, doi:10.3390/healthcare11101369_

Round 1
Reviewer 1 Report
Overall comments:
This study poses a clinically relevant question concerning resource allocation when using robot-assisted therapy. As a major comment to the study, the researchers should conform to the CONSORT checklist, to make sure that all details for allowing replication of the study, is reported. The manuscript in its present form is poorly reported, which makes it near impossible to replicate the study. Examples of necessary points which weren't addressed, are:
· Description of the setting
· Description of the intervention (Needs to be more than only three lines to allow replication)
· Determining sample size
· Description of randomization procedure
· Adherence to therapy (I would be very surprised if all subjects completed all 20 sessions)
· Full flowchart with n for excluded patients, and reason for excluding
· Trial registration
·
Minor comments.
The study was ethically approved in 2012 but data collection was not started until eight years later! What was the reason for this?
Were all 36 patients hospitalized at your department for four weeks?
Exclusion: Why were pregnancy and breest feeding regarded as exclusion criteria? Seems a bit arbitrary to me.
What does this mean: "When the robot training started in the experimental group, verbal and visual instructions using a stick were given if the therapist thought there was a problem with the patient's treatment performance. We also allowed encouragement other than objective instructions."
The FIM has 18 items, not six.
Table 1. It says 19 in number of participants in each group, but 18 in text and flow chart. Please align numbers.
Needs extensive language editing.
Author Response
I appreciate for pointing out things to correct.
I have attached the detailed corrections in the attachment.
Please see the attachment.
Sincerely.

Reviewer 2 Report
This is an excellent manuscript and a good topic to add to the literature. My comments are only minor.
All tables: Be sure that they appear on the same page so that it is easier to read.
Line 62: extra period after "Board"
Figure 1: Control group post-treatment should be pre-treatment in the middle on the right. Also, please indicate the intervention was 5 days a week in addition to the other information provided.
Figure 2: the second two pictures are redundant. It might be more helpful to have a better image of the arm in the device. You can barely see the arm and hand in the second picture.
Figure 3: Since these images are from programs (presumably not written by the authors), I would expect explicit references for all of them.
Results: I believe the results are given as mean and standard deviation; however, in tables this is written as Mean (Standard Deviation) and in the text it is written as Mean (plus minus symbol) Standard Deviation. I think the former format is clearer and would suggest be consistently used.
Table 1 and study subjects in general: This table does a nice job of comparing the two groups; however, it is odd that you have no statistical difference between the groups for ANY of the measures, given that you did not screen or control for ANY of the measures. Statistically, this seems highly unlikely that you would randomly get two groups that are so similar. Additionally, the number of patients is not consistent: 36 in the text (line 50), 19 in each group (Table 1), and a mean number of subjects in each group of 20 (also Table 1). I would also check you statistics on the Duration as you seem to have two distributions with larger standard deviations than the distribution combined (which is possible, I suppose).
Table 2: The indication of significance is confusing, especially between groups (the #). Perhaps this could be indicated differently.
Figure 4: I think all of these are significant, but the figure does not indicate that. In fact, it seems to suggest that anything significant would be marked with an *, but nothing is. This figure also needs to have units.
Discussion, paragraph 1. This paragraph might be better placed in the Introduction (without the last sentence) or the Materials and Methods section.
Author Response

(The authors gave the same response as above.)

Reviewer 3 Report
1. Have authors comments on why spasticity not altered? "authors claim no change in spasticity in both groups with intervention.
2. What is the cost and training of robotics as opposed to traditional therapy.
3. Early Intervention. What would results be if intervention delayed 3-6 months until patient stabilized
4. Postulate if therapy occurred to 6 months. Would results be the same?
1. Why spasticity not altered in both groups?
2. What is the cost for training with robots as opposed to traditional treatment
3. What would impact be if robots delayed for 6 months?
4. Does age, weight , carry over with significant other, education, vocation affect outcome?
Author Response

(The authors gave the same response as above.)
